# CryoEM structure of *Drosophila* flight muscle thick filaments at 7 Å resolution

Nadia Daneshparvar[1,2] , Dianne W Taylor[2] , Thomas S O'Leary[3] , Hamidreza Rahmani[1,2] , Fatemeh Abbasiyeganeh[2] , Michael J Previs[3] , Kenneth A Taylor[2]

Striated muscle thick filaments are composed of myosin II and several non-myosin proteins. Myosin II's long α-helical coiled-coil tail forms the dense protein backbone of filaments, whereas its N-terminal globular head containing the catalytic and actin-binding activities extends outward from the backbone. Here, we report the structure of thick filaments of the flight muscle of the fruit fly *Drosophila melanogaster* at 7 Å resolution. Its myosin tails are arranged in curved molecular crystalline layers identical to flight muscles of the giant water bug *Lethocerus indicus*. Four non-myosin densities are observed, three of which correspond to ones found in *Lethocerus*; one new density, possibly stretchin-mlck, is found on the backbone outer surface. Surprisingly, the myosin heads are disordered rather than ordered along the filament backbone. Our results show striking myosin tail similarity within flight muscle filaments of two insect orders separated by several hundred million years of evolution.

## Introduction

Sarcomeres of striated muscle are composed of four basic components: bipolar, myosin-containing thick filaments; polar, actin-containing thin filaments; a Z-disk which cross-links antiparallel actin filaments into a bipolar structure; and a connecting filament to link the thick filaments to the Z-disk. Of these four elements, the thin filaments are better characterized than the others. Here, we are concerned with the myosin-containing thick filaments, the least characterized component structurally.

Molecules of myosin II, the only filament forming myosin (Foth et al, 2006), are heterohexamers consisting of a pair of identical heavy chains of ~2,000 residues and two pairs of light chains (Fig 1A and B), dubbed essential and regulatory. Myosin's head comprises the N-terminal ~850 residues plus one of each light chain, the remaining ~1,150 residues form a continuous α-helical coiled-coil tail. Myosin heads in thick filaments of relaxed flight muscles of the large water bug *Lethocerus indicus* extend outward in intervals of

145 Å (Fig 1C) giving the appearance of a ring encircling the filament backbone, a structure dubbed a crown.

In an active muscle generating tension, individual myosin heads act as independent force generators (Huxley, 1974) and are disordered generally except when attached to actin. In relaxed muscle, in which state the muscle is easily extended because actin–myosin interactions are inhibited, myosin heads become ordered (Huxley & Brown, 1967), although details of their ordered arrangement were obscure for many years. In 2001, the relaxed (inhibited) myosin head conformation of smooth muscle myosin II was visualized (Wendt et al, 2001) and later dubbed an interacting-heads motif (IHM; Fig 1B). All relaxed thick filament structures reported since then that resolve individual myosin heads show this same conformation (Craig, 2017). High asymmetry characterizes the head–head interaction of the IHM with the actin-binding surface of one head, the blocked head, juxtaposed to the side of the other head, the free head, so named because its actin-binding interface is not blocked. In most relaxed thick filaments, the IHM lies roughly tangential to the filament backbone with the free head actin-binding surface facing the backbone, thus preventing actin binding by both heads via different mechanisms. However, in the giant water bug *Lethocerus*, the IHM lies perpendicular to the backbone (Fig 1C) with the free head bound to the filament backbone by a different mechanism producing an orientation that is so far unique in striated muscles (Hu et al, 2016). With few exceptions which occur among primitive single-cell organisms, ability to form the IHM is nearly ubiquitous for organisms expressing myosin II (Jung et al, 2008; Lee et al, 2018).

Whereas the head is critical for actomyosin motion, myosin's ~1,600 Å long tail is integral to filament formation (Fig 1A). Using proteolysis in high salt solutions where myosin is soluble, the tail cleaves into two domains, the first ~1/3rd comprising subfragment 2 (S2) and the remaining 2/3rd comprising light meromyosin (Fig 1A). S2 is soluble at physiological ionic strength, but light meromyosin is not suggesting that light meromyosin contains the thick filament assembly activity with the segment of S2 proximal to the myosin heads providing a tether enabling them to search for and bind actin subunits on the thin filament.

Muscle thick filaments from all species contain additional proteins that modulate their activity or that determine their length.

---

[1]Department of Physics, Florida State University, Tallahassee, FL, USA   [2]Institute of Molecular Biophysics, Florida State University, Tallahassee, FL, USA   [3]Department of Molecular Physiology & Biophysics, University of Vermont College of Medicine, Burlington, VT, USA

Correspondence: ktaylor@fsu.edu

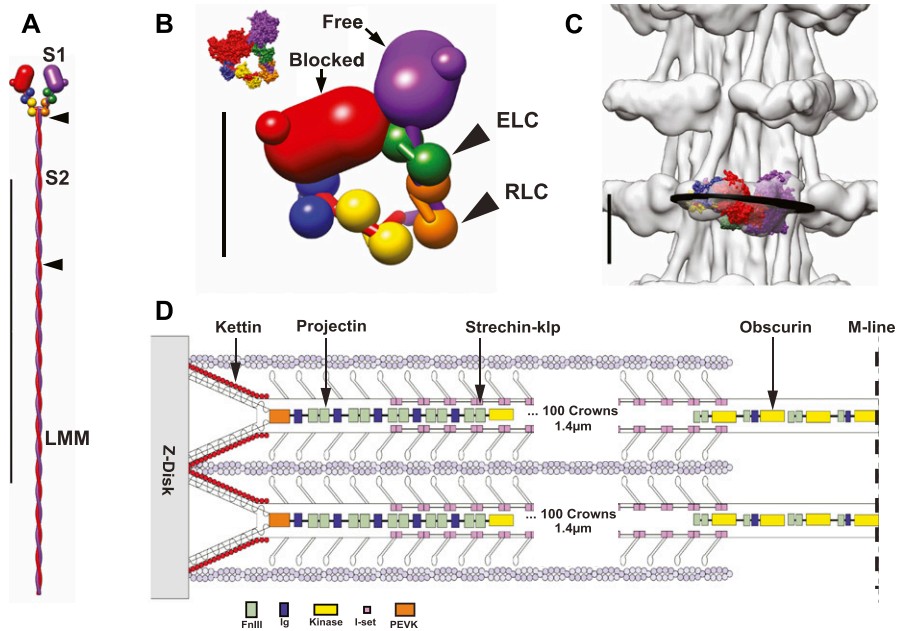

**Figure 1. Myosin filament features.**
**(A)** Diagram of a myosin molecule with two equivalent heads and an α-helical coiled-coil tail. Proteolysis at two sites (arrowheads) fragments the molecule into two separate heads (S1) and two tail segments (S2 and LMM [light meromyosin]). **(A, B, C)** Vertical line represents 1,000 Å in panel (A) and 100 Å in (B, C). **(B)** The interacting heads motif (IHM). In the IHM, the two heads are not equivalent. Instead, the actin-binding domain of one head (blocked) contacts the adjacent head (free) whose actin-binding domain is not blocked. The inset shows the space-filling structure of PDB 1I84 (Wendt et al, 2001). In filaments, the free head is usually juxtaposed to the thick filament backbone effectively preventing it from binding actin in the relaxed state. **(C)** The IHM placed within the *Lethocerus* thick filament reconstruction at 20 Å resolution. The black disk approximates the orientation of a best plane drawn through the IHM. This orientation is unique in striated muscle. **(D)** Schematic diagram showing the relative placement of the giant proteins, kettin, projectin, obscurin, and stretchin-klp within a sarcomere. Projectin binds mostly at the filament tip, obscurin to the M-band (bare zone), and kettin to the thin filament and projectin. Stretchin-klp binds along the main shaft of the thick filament but not in the bare zone or at the filament tips. **(A, B, C)** Coloring scheme in (A, B, C)—blocked head: heavy chain, red; essential light chain, blue; regulatory light chain, yellow; free head: heavy chain, purple; essential light chain, green; regulatory light chain, orange. Scale bar in (A) is 1,000 Å, in (B, C), 100 Å. **(D)** Coloring scheme for (D)—Fn3 domains, light green; Ig domains, blue; Ig domains of stretchin-klp, pink; kettin, red; obscurin kinase domains, yellow. **(D)** Adapted from Bullard et al (2005). **(A, B, C)** from Hu et al (2016).

In *Drosophila* flight muscle, two of these proteins, projectin and kettin form the connecting filament between thick filaments and the Z-disk (Fig 1D). Both are largely confined to the filament ends (Lakey et al, 1990; Ayme-Southgate & Southgate, 2006). Obscurin is a bare zone–binding protein whose length is about sufficient to reach the first crown of myosin heads (Katzemich et al, 2012). Stretchin-klp is distributed along most of the A-band (Patel & Saide, 2005). Three other proteins (not shown in Fig 1), paramyosin and miniparamyosin (Becker et al, 1992), flightin, and myofilin (Qiu et al, 2005) are located in the filament core or among the myosin tails.

Myosin tails provide more than a device for thick filament assembly; they are involved in the activation of myosin heads. Vertebrate skeletal muscle can shorten at low tension even with most myosin heads ordered as in relaxed muscle, but at high loads, the myosin heads become disordered (Linari et al, 2015). The effect, which must be manifest by a change of some kind in the thick filament backbone, has been interpreted as mechano-sensing by the thick filaments (Irving, 2017). Of the ~500 myosin mutations known to cause muscle disease in humans, ~40% are located in the tail domain (Colegrave & Peckham, 2014). Myosin tail mutations may result in incorrect assembly of thick filaments, affect the function of correctly assembled thick filaments, or affect stability resulting in increased turnover. The myosin tail amino acid sequence is highly conserved. The *Drosophila* (fruit fly) and *Lethocerus* (large water bug) flight muscle myosin tail sequences are 88% identical to each other, and when compared with human cardiac β-myosin (MYH7), the sequences are 54% identical, 74% similar.

Asynchronous flight muscle, so named because contractions are out of synchrony with the nervous stimulation, is a comparatively recent adaptation in insects (Pringle, 1981). Four insect orders use this type of flight muscle: Diptera (flies, including *Drosophila* sp.), Hemiptera (true bugs, including *Lethocerus* sp.), Hymenoptera (bees and wasps), and Coleoptera (beetles). Asynchronous flight muscles have several characteristics in common (Pringle, 1978); among them are (1) an indirect arrangement with the muscle attaching to the exoskeleton rather than directly to the wings so that contractions occur at the resonant frequency of the thorax, (2) a high degree of order in the arrangement of thick and thin filaments, (3) relatively sparse sarcoplasmic reticulum, (4) T-tubules aligned with the M-band rather than in the middle of the I-bands or with the Z-disk as occurs in vertebrate striated muscle and in synchronous insect flight muscle (Pringle, 1981), (5) a highly developed stretch activation and shortening deactivation mechanism that enables the muscle to contract rapidly at constant but submaximal calcium concentration, and (6) thick and thin filaments nearly completely overlapped at rest length with the result that the amount of shortening is quite small, being only a few percent of the sarcomere length. Stretch activation is most refined in asynchronous flight muscle but is also observed in vertebrate striated muscle, more so in cardiac than in skeletal muscle (Pringle, 1978).

Recently, a 5.5 Å 3D image of *Lethocerus* flight muscle thick filaments revealed the myosin tail packing in unprecedented detail along with several unexpected features (Hu et al, 2016). Contrary to the generally accepted model for the myosin tail packing into subfilaments (Wray, 1979), the myosin tails were found organized into curved molecular crystalline layers (Squire, 1973), referred to here as ribbons because of their rather flat and narrow but elongated morphology. Within ribbons, myosin tails from adjacent molecules are offset by 3 × 145 Å, which favors extensive tail contact within ribbons, and somewhat less contact between ribbons (Hu et al, 2016), as well as an enhanced matching of regions of complementary charges (McLachlan & Karn, 1982). Intercalated among the myosin tails were four separate densities with the morphology of extended polypeptide chains. Within the center of the filament was a set of rod-like densities, which are most likely paramyosin.

*Lethocerus* sp. have the distinct size advantage over *Drosophila* for structural studies but cannot be genetically manipulated. *Drosophila* sp. flight muscles can be genetically manipulated often without consequence to their laboratory survival and further breeding. *Drosophila* and *Lethocerus* belong to insect orders that diverged by some accounts ~373 million years ago (Misof et al, 2014). Their flight muscles have very different contraction frequencies and sarcomere lengths. Both have several non-myosin proteins in common, but with some differences in size, sequence, and quantity. Here, we report two nearly identical 7 Å resolution reconstructed images of relaxed thick filaments from two strains of *Drosophila melanogaster*, a wild-type and a regulatory light chain (RLC) mutant that differ significantly from images of *Lethocerus* in the order of the myosin heads and in the non-myosin proteins. Significantly, the myosin tail arrangements are highly similar. The muscle specific changes appear to be orchestrated around the myosin tail arrangement (ribbons) as a basic structure.

## Results

### Thick filament appearance in vitreous ice

Thick filaments were isolated from flight muscles of two *Drosophila* strains, a wild-type (WT), W1118, and a strain designated Dmlc2[Δ2–46; S66A,S67A] (Farman et al, 2009) with RLC mutations S66A and S67A plus deletion of N-terminal residues 1–46. RLC phosphorylation is known to disorder the heads (Levine et al, 1996), but would be impossible in the mutant, which made it a useful test of the possibility that phosphorylation was the agent of myosin head disorder.

Thick filaments of *Drosophila* flight muscle have a length of 3.2 μm (Gasek et al, 2016). Images recorded at a magnification of 18,000× on a DE-64 camera showed almost the entire 2-μm-diameter hole in the support film (Fig 2A and B), facilitating bare zone and filament tip identification from which filament polarity could be determined a priori (Fig S1A). When the distribution of filament segments included in the reconstruction at the beginning and at the finish is displayed as a histogram, the vast majority come from the region between crowns 0 and 60 (Fig S1B). The large number of segments including crown 0 is indicative of the frequency with which bare zones were visible.

Micrographs of frozen hydrated filaments isolated from both strains, generally appeared straight, but bent or broke usually at the bare zone in the filament center (Fig 2A and B; see also Fig S1A). Filament density was uniform across the bare zone but on either end appeared hollow all the way to the filament tip. Images from both sets of thick filaments showed disorder in the myosin heads (Fig 2A and B) in contrast to the distinct crown structure seen with *Lethocerus* thick filaments (Hu et al, 2016). The head disorder is also apparent at higher magnification (Fig 2C). After computing the

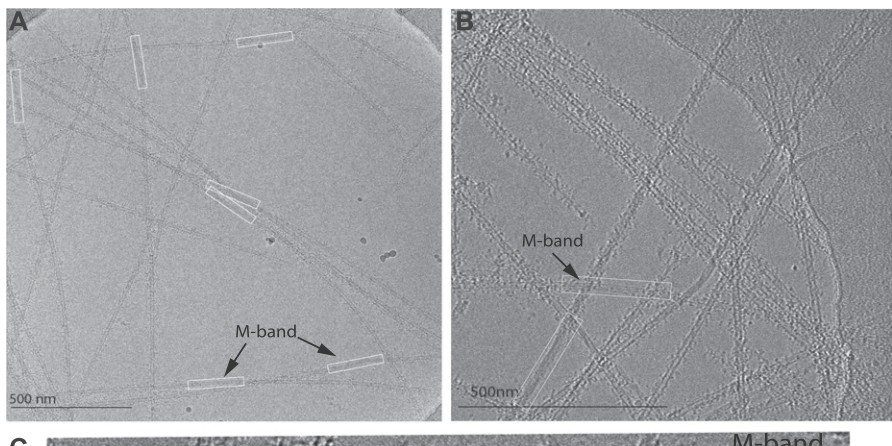

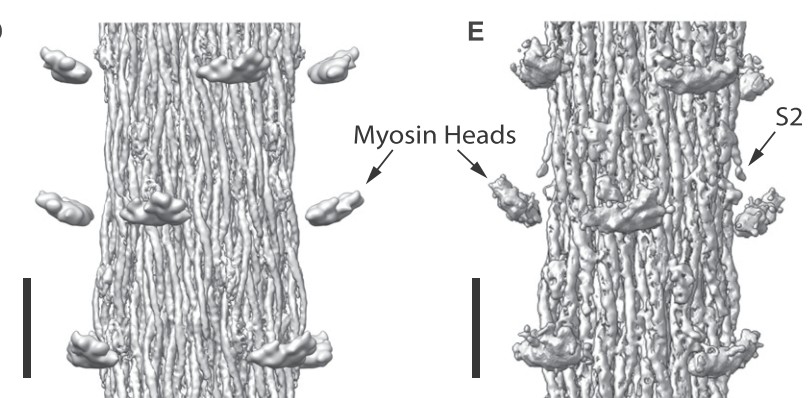

**Figure 2. Typical thick filament images and reconstructions of WT and mutant flies.**
**(A)** WT-isolated thick filaments. In the adjacent A-bands, numerous densities (myosin heads) project from the surface. **(B)** Thick filament from regulatory light chain mutant flies. Images for this set recorded using a Volta phase plate. Note that the density across the diameter of the bare zone is uniform but across the A-band is lighter in the middle, suggesting the filaments are hollow. **(C)** Image of the WT thick filament showing the disorder of the myosin heads and the smoothness of the bare zone, outlined in the white box. **(D)** WT reconstruction after imposing helical symmetry and application of local deblur. **(E)** Mutant reconstruction after similar treatment. **(D, E)** Images in panels (D, E) low-pass filtered to 7 Å resolution for the backbone and 40 Å resolution to display the floating head densities.

wild-type reconstruction, we obtained the Dmlc2[Δ2–46; S66A,S67A] flies at which point we also had access to, and used, a Volta phase plate for recording its image data.

The wild-type reconstruction showed little detail in the myosin heads, which appeared as floating densities with no visible S2 connection to the backbone (Fig 2D). Despite enhancement of the filament contrast by the phase plate (Fig 2B), no significant improvement in myosin head detail was obtained (Fig 2E).

### Reconstruction

Very little density recognizable as myosin heads was visible in both the wild-type and Dmlc2[Δ2–46; S66A,S67A] reconstructions. However, resolution of the filament backbone was ~7 Å (Figs S2 and S3) revealing the myosin tail α-helices and non-myosin proteins (Fig 2D and E). Both reconstructions are nearly indistinguishable and can be described adequately using just the wild-type result. Both wild-type and mutant had very similar helical angles of 33.86° and 33.92°, respectively. Slightly more subfragment 2 is visible in the Dmlc2[Δ2–46; S66A,S67A] reconstruction.

Our *Drosophila* thick filaments have an outer diameter of ~180 Å, slightly more than the ~170 Å predicted for a four stranded filament (Squire, 1973), but slightly less than the 190 Å observed for *Lethocerus*. A three-stranded thick filament has a predicted backbone diameter of ~155 Å (Squire, 1973). We computed reconstructions from both the wild-type and mutant using three rotational symmetries: twofold (C2), threefold (C3), and fourfold (C4). The C2 reconstruction was very similar to the C4 but of lower resolution because of less averaging with just C2 imposed; with C3 symmetry imposed, the reconstruction was distinctly different and of lower quality (Fig S4). Imposing C3 symmetry produced a reconstruction with three robust densities running along the thick filament axis with comparatively flimsy connections very different from the generally uniform cross section density found in other threefold symmetric thick filaments at comparable resolution. Generally high-quality transmission electron micrographs of transverse sections through *Drosophila* flight muscle show thick filament profiles that are rotationally uniform with little hint of symmetry (Reedy et al, 2000; Farman et al, 2009) as opposed to the highly symmetric profile seen in the reconstruction with C3 symmetry imposed. cisTEM (Grant et al, 2018) was used to compute the final C4 reconstruction. Helical parameters were then determined and imposed using RELION (Scheres, 2012), which produced a smoother map. Last, the map was sharpened using local DeBlur (Ramirez-Aportela et al, 2020). All these maps are the same in terms of resolution (Figs S2 and S3), but the sharpened map better separates the coiled-coil α-helices and non-myosin proteins.

Extensive averaging of multiple filaments and multiple repeating segments of each filament occurs in the reconstruction process. Consequently, structural elements that do not follow the myosin symmetry, that follow it but are present in less than equimolar amounts, or that are conformationally heterogeneous will not show up in the reconstruction or will only be visualized at a lower contour threshold. Those elements that appear in the reconstruction at the same contour threshold as the myosin tails are present at close to equimolar amounts with respect to myosin and are comparatively conformationally homogeneous.

### Myosin heads are disordered

Well-resolved myosin heads in a modified position of the IHM characterized the relaxed *Lethocerus* thick filament (Fig 1C). Surprisingly, no density resembling the IHM was visible in either *Drosophila* reconstruction even when the map is low-pass filtered to 40 Å (Fig 2D and E). Instead, four densities occur in the expected axial position of myosin heads but unconnected to the filament backbone and comparatively small and shapeless (Fig 2D and E), which is characteristic of a disordered structure. Isolation procedures used for *Drosophila* and *Lethocerus* thick filaments were virtually identical (Hu et al, 2016) except that the *Drosophila* thick filaments were made from either fresh tissue or following brief storage at −80°C in glycerol buffer. Light chain phosphorylation, which typically disorders the myosin heads (Levine et al, 1996), cannot be the cause because the mutant, which cannot be phosphorylated, also showed disordered heads. The putative myosin head density is axially aligned with the N terminus of the visible, ordered part of the myosin tail, the proximal S2. In *Lethocerus* thick filaments, the S2 extends ~110 Å from this point to connect the head–tail junction at the next Z-ward crown (Hu et al, 2016); it is this connection that is not visible in *Drosophila*.

### Myosin tails are arranged in ribbons

After extending the reconstruction to a length of 12 crowns, a single myosin tail could be segmented from the map (Fig 3A). Continuous density corresponding to the individual myosin tail α-helices is visible for a distance of 10 crowns; the 11th crown of myosin tail would contain the disordered proximal S2. Myosin tails are arranged within an annulus of outer diameter 180 Å surrounding a hollow core of diameter 70 Å, which is distinct from *Lethocerus* thick filaments where the central core had eight densities believed to be paramyosin. Like the *Lethocerus* thick filament (Hu et al, 2016), the *Drosophila* myosin tail extends from its N terminus on the outside of the tail annulus through to its C terminus on the inside giving an outward tilt of 1.8° relative to the filament axis.

Evidence for the validity of the ribbon arrangement of myosin tails is provided by the myosin tail packing. A transverse slab through the reconstruction always shows 40 myosin tails because of the fourfold symmetry of the filament and the 10-crown length of the myosin tail. The proximal S2 extends outside of the myosin tail annulus and is not involved in myosin tail packing within the backbone. These 40 tails are arranged in 12 ribbons. By symmetry, a three-crown length of a ribbon contains a complete myosin molecule, cut into three 3-crown lengths and one extra crown length. Thus, a transverse slab through a ribbon shows a width of three myosin tails 2/3rds of the time, and 1/3rd of the time a fourth myosin tail. Segmentation of all the myosin tails revealed a ribbon packing very similar to that found for *Lethocerus* (Fig 3B and C) with myosin tails offset by three crowns, the optimal offset predicted by the amino acid sequence (McLachlan & Karn, 1982). When a segmented *Lethocerus* ribbon is superimposed on one from *Drosophila*, there is almost complete superposition (Fig 3D and Video 1) with the exception of the disordered *Drosophila* proximal S2.

### Identification of non-myosin proteins

After segmenting the reconstruction, four groups (following the imposed fourfold symmetry) each with three non-myosin densities

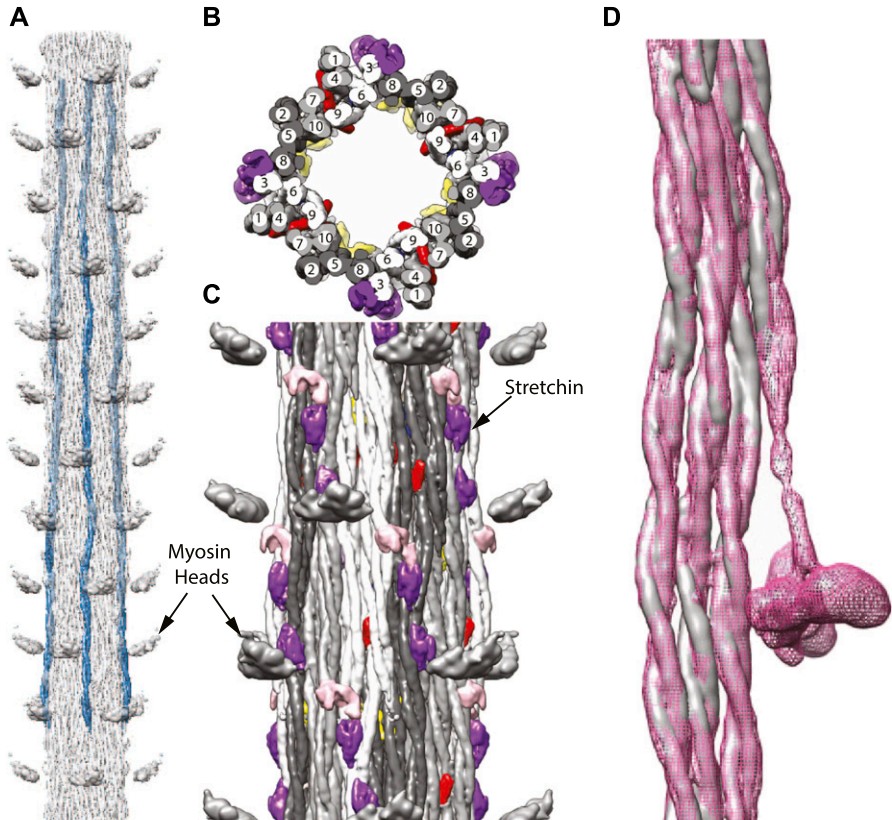

**Figure 3. Arrangement of myosin and non-myosin proteins.**
**(A)** Four symmetrically placed myosin tails (blue) segmented from a reconstruction extended 12 crowns. As in *Lethocerus* thick filaments, myosin tails run mostly parallel to the filament axis with a slight tilt inward toward the C terminus. **(B)** View looking down the filament axis showing the "curved molecular crystalline layers" (ribbons) (Squire, 1973). 10 myosin tails in each of the four asymmetric units, arising from the fourfold symmetry, are numbered sequentially according to their 145 Å axial offsets. Each ribbon consists of myosin tails offset axially by 3 × 145 Å, that is, 1, 4, 7, and 10; 2, 5, and 8; and 3, 6, and 9, starting from the point where the tail enters the backbone. Ribbons are colored white, light, and dark gray. **(C)** Longitudinal view from the outside. Non-myosin proteins flightin (red) and myofilin (yellow) are embedded among the myosin tails. A third, possibly stretchin-klp (purple and pink) is on the outer surface. **(D)** Myosin tails of *Drosophila* (gray) and *Lethocerus* (pink mesh) superimposed as a ribbon. With the exception of the proximal S2 of *Lethocerus*, which bends to the left, the same feature in *Drosophila* is mostly disordered but what is visible appears to follow a straight trajectory. Note that the density threshold for the *Lethocerus* reconstruction was chosen to be the minimum that would show the position of the free head without blurring the proximal S2.

were visible among the myosin tails (Figs 3B and C and 4A–E). On the backbone surface were four additional groups each with three non-myosin densities (Fig 4F–H). The major reported non-myosin, thick filament-associated proteins in *Drosophila* flight muscle are flightin (Vigoreaux et al, 1993), myofilin (Qiu et al, 2005), stretchin (Champagne et al, 2000; Patel & Saide, 2005), paramyosin, miniparamyosin (Maroto et al, 1996), kettin (Lakey et al, 1993), projectin (Hu et al, 1990), and obscurin (Katzemich et al, 2012). Can any of these non-myosin proteins account for these densities?

The molar abundances of non-myosin proteins in both myofibrils and isolated thick filaments were quantified by label-free mass spectrometry (MS) analyses. Isolation protocols for both myofibrils and thick filaments were nearly identical to those used in the cryoEM studies except for an additional centrifugation step to separate filaments from proteins in the supernatant. For MS, samples were digested to peptides with trypsin. The resultant peptides were separated by ultrahigh-pressure liquid chromatography (UHPLC) and directly analyzed by electrospray ionization MS (see the Materials and Methods section). The relative abundance of myosin to each non-myosin protein was determined from the liquid chromatography (LC) peak areas of the three most abundant peptides from the myosin heavy chain, essential and RLCs, and non-myosin proteins (O'Leary et al, 2019). These relative abundances were divided by 2 (except for paramyosin which is a dimer) to determine the average number of double-headed myosin molecules per non-myosin thick filament associated protein (Table 1). Although myofibrils were present in sufficient quantity for

measurement in triplicate, sufficient isolated filaments were available for only a single measurement.

Flightin, myofilin, stretchin, paramyosin, and miniparamyosin were the most abundant non-myosin thick filament associated proteins in the *Drosophila* myofibrils and the most likely to be seen in the reconstruction. Their abundance was consistent with previous biochemical observations (Beinbrech et al, 1985; Ayer & Vigoreaux, 2003; Qiu et al, 2005). Kettin, projectin, and obscurin were present at lower stoichiometries in the myofibril samples. Kettin and projectin were largely absent in the thick filament samples. Kettin in particular is susceptible to calpain cleavage (Lakey et al, 1993) which may be the agent of its loss. Kettin and projectin are mostly at the filament tips where very few segments were selected for the reconstruction (Fig S1B). They, thus, make a negligent contribution to the reconstruction. Obscurin is an M-band protein with little extension beyond the first crown of myosin heads. Although present, its contribution is heavily diluted by the overweighting of segments beyond where it could contribute.

Three non-myosin densities (red, yellow, and blue in Figs 3B and C, 4A–E, and Videos 2–Videos 4) are similar to those observed in *Lethocerus* thick filaments. One density (red) penetrates a ribbon and corresponds in location and shape to the putative flightin in *Lethocerus* thick filaments (Fig 4C and D) (Hu et al, 2016). Flightin's nearly 1:1 stoichiometry with myosin (Table 1) (Ayer & Vigoreaux, 2003; Qiu et al, 2005) is consistent with this assignment. *Drosophila*'s putative flightin density does not extend as far outside the filament backbone as in *Lethocerus* (Fig 4E), where it made a contact

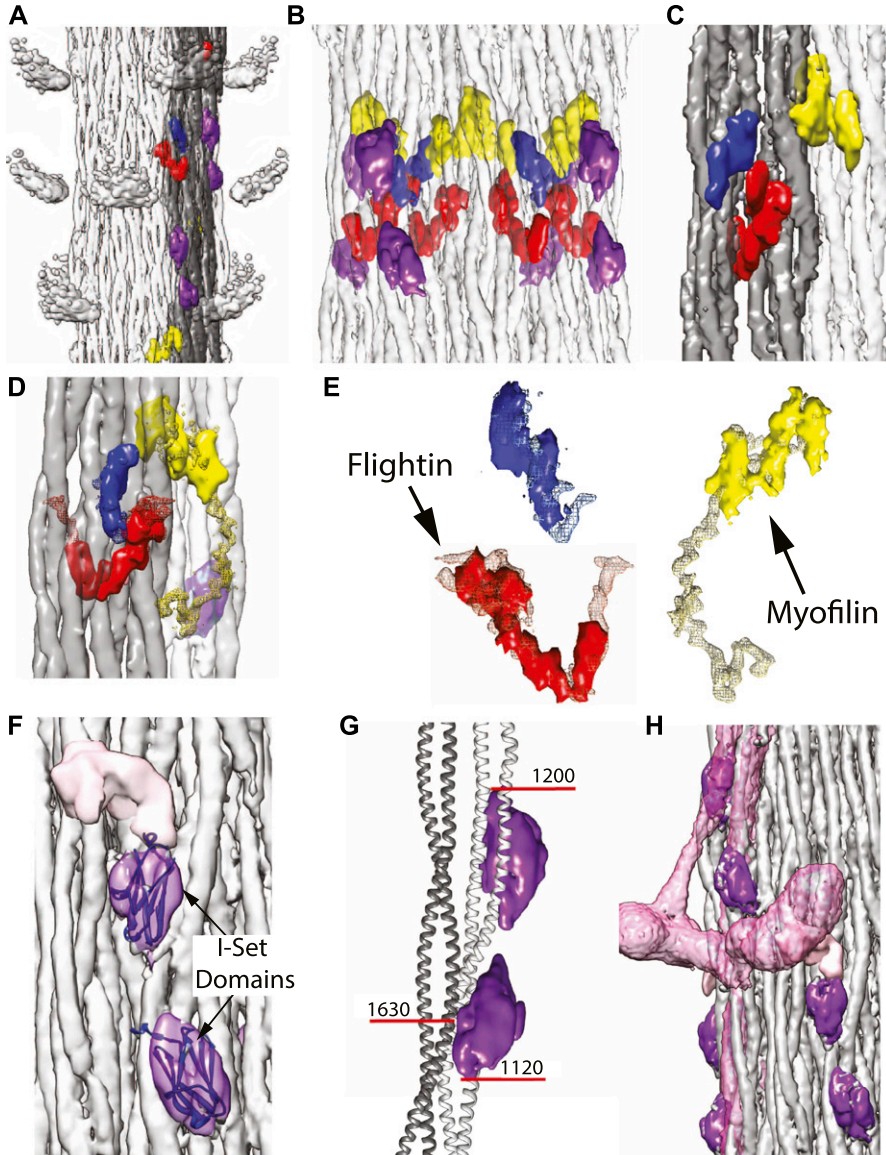

**Figure 4.  Non-myosin proteins of the *Drosophila* thick filament reconstruction.**
**(A)** View from the outside showing the myosin head density in relationship to the non-myosin proteins. **(B)** View from the outside at higher magnification looking through the myosin tails. The putative stretchin-klp densities (purple and pink) of *Drosophila*, which are not seen in *Lethocerus*, are found only on the outside of the thick filament backbone. **(C)** View from the inside looking out. Flightin (red) extends through the gray ribbon, whereas myofilin (yellow) is at the edge of the ribbon and the blue protein positioned on its surface. **(D)** The blue density binds the inner surface of the gray ribbon; the yellow (myofilin) density binds between the gray and white ribbon; the red density penetrates the gray ribbon. **(E)** Red, blue, and yellow non-myosin densities of *Drosophila* superimposed on the corresponding feature of *Lethocerus* displayed as mesh. **(F)** View from the outside showing the putative stretchin-klp density. An atomic model of an I-set domain has been built into two of the densities. The third (pink) is shown without an atomic model. Threshold for I-set/tail is 4.65 and threshold for stretchin linker is 1.00. **(G)** The paired Ig-like densities shown on a ribbon model of the myosin tail. **(H)** The putative stretchin-klp densities superimposed on a *Lethocerus* thick filament reconstruction where they appear to pass under the free head and S2. Note that the density threshold for the *Lethocerus* reconstruction was chosen to be the minimum that would show the position of the free head without blurring the proximal S2. Coloring scheme same as Fig 3. One ribbon is colored gray. **(A, B, D)** Myosin tails in panels (A, B, D) are at 50% transparency.

with the proximal S2, which is disordered in *Drosophila*, thus disordering any flightin density associated with it. A second non-myosin density held in common (yellow) was tentatively identified as myofilin in *Lethocerus* (Hu et al, 2016). Myofilin has a slightly lower stoichiometry than flightin with respect to myosin molecules (Table 1) and in *Drosophila* is significantly smaller than in *Lethocerus* (Fig 4E). The *Lethocerus* putative myofilin density had a domain close enough to paramyosin to suggest an interaction; it is this domain that is visible here even though paramyosin is not. Both putative myofilin and flightin densities in *Drosophila* are too small to contain a full molecule. A third non-myosin density (blue in Fig 4A–E and Video 4), which may be a portion of either flightin or myofilin, linked by a disordered peptide, remains unidentified in both species. Its size and shape are nearly identical to that of *Lethocerus*.

Calpain cleavage can be detected by MS as the absence of peptides after digestion. The N-terminal 64 residues of flightin were

not detected in the filament preparations but were detected in the myofibril digests. We detected no calpain cleavage of myofilin in the filament preparations. Note that previous reports of calpain cleavage of flightin and myofilin in *Lethocerus* flight muscle were obtained from Z-disk preparations, not from thick filament preparations (Bullard et al, 1990). Calpain cleavage of stretchin-klp was also detected (see below).

Three additional densities not observed in *Lethocerus* (purple and pink in Figs 3B and C, 4F–H, and Videos 2–Videos 5) were observed on the outside surface of the *Drosophila* thick filament backbone. The two purple densities were similar but not identical. At a somewhat lower contour threshold, the pink density appears, suggesting that another feature is present closely linked to the purple densities but with lower occupancy or greater heterogeneity. Based on the proteomic data, these three additional densities appear to be from stretchin-mlck's Ig-like domains and linkers.

**Table 1.** Summary of mass spectrometry results.

| Gene | Common name | Accession no. | Molecular weight (kD) | Myosin molecules per protein molecule (myofibril) | Myosin molecules per protein molecule (filaments[a]) |
|------|-------------|---------------|----------------------|---------------------------------------------------|------------------------------------------------------|
| Fln | Flightin | P35554 | 20,656 | 1.3 ± 0.4 | 1.5 ± 0.5 |
| Mf | Myofilin | Q9VFC7; C1C553 | 41,667 | 2.0 ± 0.7 | 2.0 ± 0.6 |
| strn-mlck | Stretchin-mlck | A1ZA73 | 215,065 | 3.4 ± 1.1 | 7.9 ± 2.5 |
| Prm | Paramyosin | P35415 | 102,338 | 23.5 ± 7.9 | 30.5 ± 9.6 |
| Prm | Miniparamyosin | P35416; M9NDM6 | 74,277 | 9.7 ± 3.3 | 14.7 ± 4.6 |
| Bt | Projectin | L0MN91 | 992,527 | 30 ± 10 | Not detected |
| Sls | Titin/kettin | Q9I7U4-2 | 548,598 | 34 ± 12 | Not detected |
| unc-89 | Obscurin | A8DYP0 | 475,000 | 42 ± 14 | 62 ± 20 |

[a]Sufficient specimen for only a single measurement. Uncertainties were determined from variability between the relative ratios generated using peptides from the myosin heavy chain, essential, and regulatory light chains.

In both the filaments and the myofibrils, 15 peptide fragments from the trypsin digests are found covering residues 128–1843 of A1ZA73. A shorter transcript of stretchin-mlck, pC1, has been reported (Patel & Saide, 2005) consisting of 711 residues, contained entirely within A1ZA73, residues 232–940 with the exception of two serines added at positions 1 and 2. The first A1ZA73 fragment overlapping pC1 occurs at residues 458–465, so no observed peptide contained the N-terminal serines of pC1. The full-length A1ZA73 appears to be expressed in both our myofibril and filament preparations.

Stretchin-mlck is a large gene from which seven potential transcripts have been predicted (Champagne et al, 2000). One of these is a kettin-like protein, referred to as stretchin-klp (Patel & Saide, 2005), which is the form detected in our proteomics. Stretchin-klp (Uniprot, isoform R, A1ZA73) consists of an unstructured N-terminal 455 residue followed by five repeats of Ig-like – short linker – Ig-like – long linker. Short linkers are 11–27 residues long and long linkers are 61–174 residues long. Full-length stretchin-klp is detected in myofibrils, but in our filament preparations, the first 455 residues are not detected, presumably clipped by calpain. Assuming that the variable long linkers form different folded structures, a sequence of five 3-domain repeats can be formed. Some servers, for example, PubMed, predict six 3-domain repeats. Regardless of five or six repeats, the predicted pattern is a pair of Ig-like repeats separated by a short linker and linked to successive pairs by a long linker. Because each repeat corresponds to one myosin dimer, the expected ratio of myosin:stretchin-klp is five (or six), close to the ratio found in the myofibril preparation, 3.4 ± 1.1 and in the filaments 7.9 ± 2.5.

Each purple density is fit well by the atomic structure of an I-set domain of myosin binding protein C (PDB 2YXM; Fig 4D), which is a type of Ig domain found in many striated muscle proteins and predicted for stretchin-klp (Video 4). Both stretchin-klp Ig domains are positioned on or near the Skip 1 region of one myosin tail on the backbone surface (Fig 4E and Video 6), identifiable by the parallel (uncoiled) α-helices characteristic of the Skip 1 region (Taylor et al, 2015). Using the distance along the myosin tail density as a measure of residue number, the contact site on one myosin tail for both Ig-like domains falls between residues 1,120–1,200 (Fig 4E). One of the Ig-like domains appears to contact a

myosin tail from an adjoining ribbon centered approximately on residue 1,630. Chains of stretchin-klp molecules could follow either left-handed or right-handed helical tracks with the shorter distance between three-domain repeats being along left-handed tracks (Figs 4F and S5A).

Was paramyosin present in the thick filament reconstructions? In *Drosophila* myofibrils, the ratio of myosin to paramyosin, 23.5 ± 7.9:1 (Table 1), was lower than the ratio for isolated filaments, 30.5 ± 9.6. Both ratios exceed the 15.4:1 ratio determined from *Drosophila* flight muscle (Beinbrech et al, 1985), which is twice the ratio of myosin heavy chain to paramyosin (8.2:1, 7.7:1) for *Lethocerus* thick filaments (Bullard et al, 1973; Levine et al, 1976). Although a putative paramyosin was visible in *Lethocerus* thick filaments, its visualization required that the contour threshold be lowered because it did not appear to follow the myosin helical symmetry. Thus, the lower paramyosin content in *Drosophila* thick filaments could account for its invisibility when the map is contoured at thresholds suitable for visualizing the myosin tails.

### Visibility of myosin heads

Our *Drosophila* structure shows no well-resolved heads. To find an explanation, we aligned a ribbon segmented from the *Lethocerus* reconstruction (EMD-3301) with our *Drosophila* reconstruction with the result that one of the *Drosophila* putative stretchin-klp domains was positioned beneath the presumptive location of the proximal S2 region and the other one positioned close enough to the free head that it could sterically prevent binding to the filament backbone (Figs 4H and S5B and C). Thus, it would seem that the purple and pink densities, which MS indicates are stretchin-klp, are preventing formation of an ordered IHM similar to that of *Lethocerus*.

## Discussion

Thick filaments from invertebrate striated muscles are ideal specimens for probing the arrangement of myosin molecules as

well as accessory proteins because the filaments are helical in structure over extended lengths. It is no surprise that so far the highest resolution thick filament structures have come from invertebrate thick filaments such as the large water bug *L. indicus* (Hu et al, 2016) and tarantula (Yang et al, 2016). Invertebrate thick filaments are highly variable between species, offering possibilities for comparative studies.

The primary disadvantage of water bugs and tarantula is that neither comes from a genetic model organism, which inhibits production of new modified strains for hypothesis testing. Many mutations of *Drosophila* flight muscle thick filaments exist. Flight muscle thick filaments from both *Drosophila* and *Lethocerus* possesses several similarities but also had some surprising differences.

### Why are myosin heads disordered

Even partially ordered myosin heads could not be obtained using procedures that worked well for *Lethocerus* thick filaments. Within regions from which myosin heads project, the filaments appear well preserved and generally straight. Isolation procedures were the same for both species but were performed on much fresher *Drosophila* muscle. Conceivably prolonged glycerination or calpain treatment resulted in loss of components from *Lethocerus* filaments, but it is equally likely that *Lethocerus* does not express a stretchin-klp ortholog.

Early efforts by others to preserve thick filaments from *Lethocerus* in vitreous ice using completely different specimen preparation methods showed well-ordered crowns of myosin heads, but disordered heads with *Drosophila* thick filaments (Menetret et al, 1990). Our methods handle the isolated filaments as little as possible to preserve ordering of the myosin heads. Thus, it seems unlikely that the specimen preparation protocols used here are responsible for the differences.

RLC phosphorylation is known to disorder myosin heads in vertebrate thick filaments (Levine et al, 1996). However, the same isolation procedures performed on the mutant fly strain Dmlc2$^{[\Delta 2-46;\ S66A,S67A]}$, whose RLC phosphorylation sites and N-terminal 46 residues had been removed (Farman et al, 2009), produced no better head ordering than the wild type. Thus, RLC phosphorylation was not the cause of the head disorder. Within intact muscle fibers, the Dmlc2$^{[\Delta 2-46;\ S66A,S67A]}$ mutant shows less order than the wild type, possibly caused by the inability of the myosin heads to bind thin filament via the RLC N-terminal extension (Farman et al, 2009). Absent thin filaments, wild-type, and mutant isolated thick filaments were indistinguishable.

*Drosophila* filaments have much less paramyosin in their core than *Lethocerus* filaments which might make them more fragile. However, broken filaments are relatively infrequent in both preparations and are usually avoided. Filaments bent at the bare zone are included in the data analysis; those bent in the A-band are not. Although this possibility cannot be excluded at this time, we think low paramyosin content is an unlikely cause of the head disordering.

Do myosin heads in relaxed *Drosophila* thick filaments form an IHM? If not, they would be the first striated muscle system that does not, as all other relaxed thick filament structures have shown this inactive myosin motif (Hu et al, 2016; Craig, 2017). Relaxed

*Lethocerus* thick filaments also have the IHM, but in an altered orientation that does not involve subfragment 2. Instead of the blocked head binding subfragment 2 and holding the free head actin-binding interface against the filament backbone, free head binding to the filament backbone in *Lethocerus* is independent of the blocked head, which binds only the free head (Hu et al, 2016). *Drosophila* flight muscle myosin in solution forms the IHM (Lee et al, 2018), but if it forms in filaments, it may be disordered because of amino acid substitutions that eliminate key interactions between myosin heads and myosin tails such as those that occur with the *Lethocerus* free head. The flight muscle myosin tail sequences of *Lethocerus* and *Drosophila* myosin are 88% identical and 98% similar, so this would seem unlikely.

*Drosophila* thick filaments have protein densities on their surface that are absent in *Lethocerus* and which could provide a steric block to free head binding. Stretchin-klp, which consists of a string of 10 (or 12) Ig domains appears to be the most likely candidate. Stretchin-klp has been studied in *Drosophila* flight muscle using antibodies, which showed labeling from a location 0.1 $\mu m$ from the M-line for a distance of ~1.2 $\mu m$ (Patel & Saide, 2005), a region that closely approximates the part included in our reconstruction (Fig S1B). Proteomic analysis indicates that the full-length form is present in the myofibrils, but in the isolated thick filaments, the first 455 residues were missing leaving only the Ig domains and their linkers. Putative stretchin-klp densities follow a left-handed helical path that passes under the location where the free head of the *Lethocerus* IHM would be positioned in relaxed thick filaments (Figs 4D, S5A and B, and Video 6). Thus, a steric block to free head binding on the thick filament surface such as occurs in *Lethocerus* seems the most likely explanation for the lack of myosin head order in *Drosophila*. The blocked head and free head could still interact, but lacking a binding site on the thick filament backbone or S2, the IHM, if present, would be disordered.

### Comparison with *Lethocerus* thick filaments

Thick filament backbones of *Lethocerus* and *Drosophila* show both similarities and differences. No density assignable to paramyosin is observed. Density putatively assigned to paramyosin in *Lethocerus* was low compared to myosin and probably does not represent accurately the arrangement of paramyosin molecules (Hu et al, 2016). *Drosophila* has about half the paramyosin per myosin molecule (1:15) as *Lethocerus* (1:7) which would explain its invisibility in the reconstruction (Table 2). *Lethocerus* thick filaments had an additional non-myosin density (colored green in Hu et al [2016]) that now appears to be the non-helical myosin C terminus of the blocked head heavy chain (Rahmani, in preparation) but is possibly disordered in *Drosophila*.

Three non-myosin densities in *Drosophila* are similar to ones previously found in *Lethocerus* (Hu et al, 2016). *Drosophila* thick filaments have a V-shaped feature, colored red here and tentatively identified as flightin, which corresponds closely in shape to the similar density seen in *Lethocerus*, including its penetration through a ribbon and a folded globular domain of similar shape and position on the inner surface. Only the part contacting the proximal S2 outside the backbone is missing here either because the proximal S2 itself is disordered or it was clipped by calpain.

**Table 2.  Comparison of *Lethocerus* and *Drosophila* thick filament proteins.**

| | | *Lethocerus indicus* | | *Drosophila melanogaster* | | Reference |
|---|---|---|---|---|---|---|
| Filament length | | 2.3 μm | | 3.2 μm | | Reedy (1967) and Gasek et al (2016) |
| Rotational symmetry | | C4 | | C4 | | Morris et al (1991), Reedy et al (1981), and this work |
| Helical parameters | | 145 Å, 33.98° | | 145 Å, 33.86° | | Hu et al (2016), Irving and Maughan (2000), Perz-Edwards et al (2011), and this work |
| Non-myosin proteins | Protein | Molecular weight | Ratio to Mhc | Molecular weight | Ratio to Mhc | |
| | Flightin | 19 kD | 1:2 | 20 kD | 1:2 | Qiu et al (2005) |
| | Myofilin | 30 kD | 1:2 | 20 kD | 1:2 | Qiu et al (2005) |
| | Paramyosin | 107 kD | 1:7[a] | 102 kD | 1:15 | Bullard et al (1973), Vinos et al (1991), Becker et al (1992) |
| | Miniparamyosin | 62 kD | | 55 kD | | Becker et al (1992), Maroto et al (1996) |
| | Projectin | 800 kD | | 800–1,000 kD | | Lakey et al (1990), Ayme-Southgate and Southgate (2006) |
| | Kettin | 700 kD | | 540 kD | | Lakey et al (1993), Bullard et al (2006) |
| | Stretchin-klp | | | 225, 231 kD | | Patel and Saide (2005) |

[a]*Lethocerus cordofanus* and *Lethocerus maximus.*

*Drosophila* and *Lethocerus* flightin are similar in size (Table 2) and have regions of sequence in common, although the entire sequences are only 35% identical (Qiu et al, 2005) compared to 88% identical in the myosin tail. Among the differences are their hydrophobicity, number of isoforms, and phosphovariants (11 isoforms in *Drosophila*, 3 in *Lethocerus*; 9 phosphovariants in *Drosophila*, 0 in *Lethocerus*). Hydrophobicity in *Drosophila* flightin is low in contrast to high hydrophobicity in *Lethocerus* (Barton & Vigoreaux, 2006). Flightin is sensitive to calpain cleavage (Bullard et al, 1990) which was also observed in our preparations.

*Lethocerus* thick filaments had a second density, also present in *Drosophila*, tentatively identified as myofilin (yellow in Figs 3 and 4). *Drosophila* myofilin is ~10 kD smaller than its *Lethocerus* ortholog and also has a smaller density visible in the reconstruction. In *Lethocerus*, the myofilin density has a globular core that might contact paramyosin and an extended domain juxtaposed to the myosin tail annulus. Only the globular core is seen in *Drosophila*; the extended domain is missing (Fig 4E). The myofilin volume resolved in both *Drosophila* and *Lethocerus* is less than the molecular weight suggested; other parts remain to be resolved or are disordered.

The blue density here is nearly identical in shape to the similar density in *Lethocerus* (Fig 4E). It is the only non-myosin density that appears nearly identical in both reconstructions, but which has yet to be even tentatively identified, although it could be part of either flightin or myofilin.

Similarity would be expected given the high sequence homology in the coiled-coil tail, but seems surprising given the differences observed in other locations of the reconstruction. High sequence homology characterizes the myosin coiled-coil tail domains: >88% identity among the four insect orders having asynchronous, indirect flight muscles; 60–75% identity among tarantula, horseshoe crab, and scallop; three species for which thick filament reconstructions have been published; and 50–54% identity between *Drosophila*

flight muscle and human striated muscle (Hu et al, 2016). Differences between *Drosophila* and *Lethocerus* flight muscle thick filaments appear in the myosin-associated proteins and the myosin head order, not in the myosin tail arrangement.

*Drosophila* and *Lethocerus* flight muscles differ in structure and performance, including (1) *Drosophila* wing beat frequency at 200 Hz is ~5× faster than *Lethocerus* (Dickinson et al, 2005); (2) *Drosophila* flight muscle myosin actin activated ATPase is correspondingly faster than *Lethocerus'* (Swank et al, 2006); (3) *Drosophila* thick filaments are 40% longer than *Lethocerus'* and have half as much paramyosin; (4) adjacent thick filaments in *Drosophila* flight muscle are axially staggered by 145 Å/3 to produce a superlattice (Squire et al, 2006), whereas *Lethocerus* flight muscle has no such superlattice (Schmitz et al, 1994); and (5) if *Drosophila* forms an IHM in relaxed filaments, an unanswered question, it is disordered.

A remarkable parallel appears between the consistency of the ribbon as a structural building block of thick filaments and the consistency of the actin subunit as the structural building block of thin filament. Thin filaments across most if not all striated muscles have in common very similar F-actin structures (Galkin et al, 2002), highly conserved because of actin's high sequence conservation, differing basically by small, although not insignificant, changes in helical angle. They differ mainly in the F-actin–binding proteins, in particular troponin and tropomyosin, changes in which produce the different functional properties (Bullard & Pastore, 2019). Hemiptera, of which *Lethocerus* is a member, are thought to have appeared in the Devonian period ~373 million years ago (Misof et al, 2014). Diptera of which *Drosophila* is a member appeared later, in the early Triassic, about 245 million years ago. Asynchronous flight muscle is thought to have evolved independently a number of times (Pringle, 1978, 1981), so it is not certain that Diptera and Hemiptera had a common ancestor with asynchronous flight muscle. Possibly more remarkable would be the independent

evolution to such a similar ribbon arrangement of myosin tails and similar helical angle. Since the Hemiptera and Diptera appeared so long ago, we expect and observe distinct differences in their flight muscle thick filaments. However, the myosin tail structure and arrangement in ribbons are nearly identical, implying a strong evolutionary pressure toward maintenance. Other features, such as the associated proteins are very different suggesting adaptation to physiological requirements, are affected by the non-myosin proteins. In this context, the myosin head, which can evolve separately from the tail, is a myosin tail associated protein.

Invertebrates appeared long before vertebrates yet even among highly different species as *Placopecten magellanicus* (scallop) and *Drosophila* the sequence identity within the myosin rod is as high as 75%. Vertebrates appeared some 480–360 million years ago by some estimates (Sallan et al, 2018), yet vertebrate thick filaments from different species are highly similar in structure at least to the extent that comparative studies have been carried out. For example, they have similar length, same rotational symmetry, have titin to determine their length, and an axial repeat of 429 Å (3 × 143 Å). It is, perhaps, worth speculating that the curved molecular crystalline layers (ribbons) first proposed in 1973 (Squire, 1973) may be far more similar to each other across species and muscle types than suggested by the myosin tail sequence identity. X-ray fiber diffraction studies of vertebrate striated muscle showed a clear preference for the ribbon structure over other models based on subfilaments (Chew & Squire, 1995). High conservation of amino acid sequence from many striated muscle myosin tails, particularly for the charged residues important for filament formation (McLachlan & Karn, 1982), suggest that the ribbons may be nearly identical among striated muscle myosin filaments. Differences among the filaments would be accounted for by small changes in the arrangement of ribbons to produce different rotational and helical symmetries accompanied by significant changes in the non-myosin proteins as adaptations to differing physiological requirements.

# Materials and Methods

Calcium-insensitive human plasma gelsolin (residues 25–406) was cloned and overexpressed in *Escherichia coli* BL21 (DE3) strain in lysogeny broth medium. The expression vector was obtained from Dr. Margaret Briggs (Duke University Medical Center). The detailed protocol is included in Supplemental Data 1.

### Thick filament preparation

For a typical preparation, indirect flight muscle is dissected from the thorax of 10 *Drosophila* flies. Typically, flies are immobilized with $CO_2$ and then dissected by removing the heads, abdomen, wings, and legs leaving only the thorax. The vast majority of protein in the thorax is indirect flight muscle. We made no attempt to separate the dorsal longitudinal from the dorsal ventral muscle or any of the nonfibrillar muscles, which are much smaller and present in much lower quantity. Further details are in Supplemental Data 1.

### Electron microscopy: grid preparation

Thick filament preparations varied in concentration. Rather than centrifuging them to increase their concentration, we varied the number of drops of suspension applied to the grids, blotting away the excess from the opposite side where the drop was applied. Before proceeding to freezing, the concentration was evaluated by preparing a negatively stained specimen exactly as that for freezing except that a final wash in 1% uranyl acetate negative stain was performed, the grids air dried, and examined in a CM120 electron microscope rather than being plunge-frozen.

Frozen specimens were prepared on plasma-cleaned 200-mesh Cu Quantifoil R2/1 reticulated carbon grids using the back-blotting technique (Toyoshima, 1989). The concentration of filaments was relatively low in these preparations. By using the back-blotting technique with reticulated carbon grids, the carbon film acts as a sieve to trap several filaments over the 2-$\mu$m diameter holes often unobstructed by crossing filaments (Fig 2A and B). The grids were then plunge-frozen in liquid ethane, cooled by liquid nitrogen, using a home-built plunge-freeze device mounted in a 4° cold room.

The grid prepared in this manner has relatively uniform ice thickness with some areas of thicker ice toward the center of the grid square with thinner areas around the perimeter. We picked holes for data collection with medium levels of ice thickness for imaging to avoid the possibility that contact with the air–water interface could be the agent of head disorder.

### Electron microscopy: data collection

The wild-type sample grids were imaged on an FEI Titan Krios transmission electron microscope operated at 300 kV. Grids were maintained at liquid nitrogen temperatures at all times after preparation. Images were recorded on a Direct Electron Ltd DE-64 camera operated in the integrating mode with 44 frames collected with a total electron dose of 26 e$^-$/Å$^2$.

Image data were recorded a DE-64 direct electron detector, which has a frame size of 8,192 × 8,192 pixels. A total of 1,510 images were collected at 18,000× magnification (2.07 Å/pix) with a total exposure of 26 e$^-$/Å$^2$ and a defocus range of −4 to −6 $\mu$m.

The second set of data was collected for the mutant Dmlc2$^{[\Delta 2-46; S66A,S67A]}$ using the Volta phase plate and DE-64 in the integration mode, which had recently become available to us. A total of 2,002 images were collected at 29,000× magnification (1.29 Å/pix) with a total dose of 26 e$^-$/Å$^2$, 21 frames, and defocus range of −0.2 to −1.0 $\mu$m (Fig 2B).

### Data analysis

After frame alignment and contrast transfer function (CTF) correction, we manually picked filaments using Appion manual picker and used RELION (He & Scheres, 2017) helical extraction to obtain the particle stack. We started with 60,000 segments with a box size of 324 × 324 pixels, which contains approximately four axial repeats of length 145 Å. The 145 Å repeat is generally referred to as a "crown" (Taylor et al, 1984). In addition to a generally low signal-to-noise ratio, the myosin heads in our thick filaments are disordered, which

could throw off the alignment significantly, unless other features within the backbone can define the axial repeat period. Initial 2D classification into 50 classes by RELION was highly inconclusive. The averages were featureless which showed the program's inability to align particles and classify them effectively. To obtain better definition among the classes, we tested ROME 1.1 (Wu et al, 2017) for 2D classification. ROME 1.1, which uses statistical manifold learning (SML), could not handle a large unbinned data set. Therefore, we binned the stack by three (box size of 128). We first ran the utility *rome_map* with 50 classes and started to see features in the backbone even though density corresponding to myosin heads remained unclear. After selecting the classes showing the clearest backbone features, we ran the utility *rome_sml* requesting 300 classes. SML turned out to be a powerful classification tool for these data. After removing non-sensical classes, ~149,000 segments remained.

Unfortunately, multiple efforts at 3D classification in both RELION and ROME failed. We then imported the best classes obtained from *rome_sml* and reprocessed them using cisTEM 1.0 (Grant et al, 2018). Using the cisTEM autorefine utility for 15 cycles and a reference obtained from the existing 5.5 Å map of *Lethocerus* thick filaments, EMD-3301, low-pass filtered to 60 Å resolution, we obtained an initial ~8 Å resolution map.

To further improve the map, we started from the beginning and extracted particles with a larger box size of 432 × 432 pixels, which is large enough to include approximately five crowns. Whole frame CTF correction was redone using GPU accelerated contrast transfer function calculation package (GCTF) (Zhang, 2016) with a particle count of 198,341. Later, the whole-frame CTF correction was replaced with a local CTF correction using GCTF. Using this enlarged data set, we repeated 2D classification using *rome_map* and *rome_sml* and selected out 148,572 good segments.

We then returned to cisTEM to perform alignment without classification then repeated the 2D classification to remove more bad particles. This cycle was continued until there was little to no room for improvement, and we got our latest map which is ~7 Å. We used the earlier 8 Å *Drosophila* reconstruction low-pass filtered to 50 Å for this reference. We found 3D classification not very helpful in the final reconstruction phase because we obtained little separation of segments into different classes, and more than 87% percent of segments would go to one class. Finally, we rechecked the helical parameters using RELION 1.4, and we obtained 148.82 Å for the helical rise and +33.86° for the helical twist. Because of the magnification error, helical rise did not match the known axial rise of 145 Å which was resolved by rescaling the pixel size from 2.074 to 2.0207.

The major 3D reconstruction challenge was our inability to align segments without a starting reference. RELION 1.4 used initially was capable of enforcing helical symmetry and would have been the preferred method. However, with the myosin heads disordered, we were unable to obtain an alignment to the 145 Å axial period. To help the alignment, we low-pass filtered the previously determined 3D structure of *Lethocerus* thick filaments to 65 Å and used it as reference. At 65 Å resolution, the non-myosin proteins are not individually resolved in the filament backbone. However, because they tend to cluster between the levels of myosin heads, they may provide sufficient density variation along the filament backbone to align the 145 Å axial period. *Lethocerus* has the same axial rise as *Drosophila* according to X-ray fiber diffraction (Irving, 2006). The helical angle of *Drosophila* has not previously been reported.

Switching to cisTEM for the reconstruction, the version of which was not capable of performing iterative helical real-space reconstruction (Egelman, 2000), meant sacrificing inclusion of the helical parameters during the reconstruction process. Nevertheless, we obtained the first 3D map using alignment to an initial reference while enforcing only C4 symmetry (Fig 2C). Enforcing helical symmetry tended to smooth out the rod structure.

The rotational symmetry of *Drosophila* flight muscle thick filaments has not been reported, so we reconstructed the two filament data sets using C2, C3, and C4 rotational symmetry. The C2 reconstruction appeared to have fourfold symmetry, whereas the C3 reconstruction looked like a three bladed screw (Fig S3).

To determine the helical rise and twist in the cisTEM reconstruction, we used the *relion_helix_toolbox* in RELION and obtained a helical rise of 148.826 Å and helical twist 33.86°. The larger value than that of the axial repeat measured by X-ray fiber diffraction (Irving & Maughan, 2000; Irving, 2006) is comparable to the difference observed for the *Lethocerus* thick filaments and attributable to magnification error. Then we moved to impose the helical symmetry on the map and compared it with the original 3D structure, and other than it, being smoother there seems to be no major difference between the two structures. For local resolution estimation, we used Mono Res, and the last step was to sharpen the map using the local deblur embedded in Scipion.

The method used for the mutant data was very similar to what we have done before. Appion manual picker was used to pick filaments and 131,658 segments were extracted using RELION 1.4 helical extraction. Using cisTEM 2D classification, we were left with 44,201 segments for 3D refinement which resulted in an ~8 Å resolution 3D structure. Helical twist for this structure is 33.92 and rise of 150.0 Å is reported using relion_helix_toolbox. By rescaling pixel size from 1.29 to 1.24, helical rise will be 145 Å which is the value we expect based on X-ray fiber diffraction.

## Map interpretation and segmentation

An individual myosin molecule is 11 crowns in length or 1,600 Å in length (11 × 145 Å). To obtain a density map of a single myosin rod structure as well as determine the rod arrangement within ribbons, extension of the reconstruction to a length of 11 crowns was required, which was carried out using the helical symmetry determined by RELION. Individual myosin rods and ribbons were segmented using UCSF Chimera. Because the myosin heads and the proximal S2 were disordered, we only observed the 10 crowns of rod tightly held within the thick filament backbone. After segmenting the rods, we were able to identify and isolate non-myosin proteins. Myosin heads were more challenging because they were barely visible at 8 Å resolution. To see them, we low-pass filtered the map to 40 Å. As shown in (Fig 3B and C), S2 is not resolved so the connection between the heads and the backbone is missing.

## Quantification of protein stoichiometry: liquid chromatography MS

Myofibril and thick filament samples were solubilized in 150 µl 0.1% RapiGest SF Surfactant (Waters Corporation) in 1.5 ml microcentrifuge tubes (50°C, 1 h). Proteins were reduced by addition of

0.75 μl of 1M dithiothreitol to each tube and heating (100°C, 10 min). Cysteines were alkylated by addition of 22.5 μl of 100 mM iodoacetamide in 50 mM ammonium bicarbonate and incubation in the dark (22°C, 30 min). Proteins were digested to peptides by addition of 25 μl of 50 mM ammonium bicarbonate containing 5 μg of trypsin (Promega) and then by incubation (37°C, 18 h). The samples were dried down in a speed vacuum device. Trypsin was deactivated and RapiGest was cleaved by addition of 100 μl of 7% formic acid in 50 mM ammonium bicarbonate and heating (37°C, 1 h). The resultant peptides were dried down. RapiGest was cleaved again by addition of 100 μl of 0.1% trifluoroacetic acid and heating (37°C, 1 h). The resultant peptides were dried down and reconstituted a final time in 60 μl of 0.1% trifluoroacetic acid. The tubes were centrifuged at 18,800 RCF for 5 min (Sorvall Legend Micro 21R; Thermo Fisher Scientific) to pellet the surfactant. The top 57 μl of solution was transferred into an MS analysis vial.

A 20-μl aliquot of each sample was injected onto an Acquity UPLC HSS T3 column (100 Å, 1.8 μm, 1 × 150 mm) (Waters Corporation) attached to an UltiMate 3000 UHPLC system (Dionex). The peptides were separated, and the UHPLC effluent was directly infused into a Q Exactive Hybrid Quadrupole-Orbitrap mass spectrometer through an electrospray ionization source (Thermo Fisher Scientific). Data were collected in data-dependent MS/MS mode with the top five most abundant ions being selected for fragmentation.

Peptides were identified from the resultant MS/MS spectra by searching against a *Drosophila* proteome database downloaded from UniProt using SEQUEST in the Proteome Discoverer 2.2 (PD 2.2) software package (Thermo Fisher Scientific). The potential loss of methionine from the N terminus of each protein (−131.20 kD), the loss of methionine with addition of acetylation (−89.16 kD), addition of carbamidomethyl (C; 57.02 kD), oxidation (M, P; 15.99 kD: M; 32.00 kD), and phosphorylation (S, T, Y; 79.98 kD) were accounted for by variable mass changes. All of the protein matches from the peptides identified are presented in Table 1. The detailed list of peptides from the myofibril preparation are provided in Supplemental Data 2. The detailed peptide list from the filament preparation is provided in Supplemental Data 3.

For the proteins reported in Table 1, the peptide sequences identified in the SEQUEST analysis were imported into Protein BLAST (https://blast.ncbi.nlm.nih.gov/) to refine the protein identification. For titin/kettin, all of the peptides were associated with the smaller 548,598 kD, Q917U4-2 isoform. For projectin, large portions of the L0MN91 sequence, spanning amino acids 837–1727 and 2,429–3,131, were not found. This may be indicative of the presence of a shorter splice variant, but not previously reported for projectin.

For label-free quantification, the Minora Feature Detector (PD 2.2) was used to identify LC peaks with the exact mass, charge states, elution time, and isotope pattern as the SEQUEST derived peptide spectral matches across the samples in the entire study. Areas under each LC peak were calculated and reported in PD 2.2 as peptide abundances. Peptide abundances were exported to Excel (Microsoft). Relative protein abundances were estimated using a label-free approach which partially mitigates differences in each peptide's ionization efficiency (O'Leary et al, 2019).

Specifically, an average abundance was determined from the LC peak areas of the top three ionizing peptides from each protein of interest. These proteins included myosin heavy chain, essential light chain, and RLC; the thick filament-associated proteins flightin, myofilin, stretchin-mlck, paramyosin miniparamyosin, projectin, titin/kettin, and obscurin. The relative stoichiometry between myosin and each thick filament–associated protein was determined from these average abundances. Three separate relative abundances were determined for each myosin-associated protein from the LC peaks of the myosin heavy chain and two myosin light chains, and SD were determined. Each myosin molecule is arranged as a dimer, consisting of two heavy chains and four light chains. With exception of paramyosin, which also exists as a dimer, each myosin associated protein exists as a monomer. To account for these structural arrangements, the relative abundances were divided by 2, where appropriate, and the number of myosin molecules per myosin-associated protein ± SD was reported.

## Data Availability

The reconstruction volumes have been deposited in the Electron Microscopy Data Base under accession codes EMD-22217 and EMD-22218. The electron microscopy data consisting of raw frames and frame-aligned images as well as metadata are deposited in EMPIAR under accession code EMPIAR-10436. The raw MS data are available in the following site: ftp://massive.ucsd.edu/MSV000085627/.

## Supplementary Information

## Acknowledgements

We thank Dr. Wu-Min Deng of Florida State University and the members of his laboratory, particularly Yi-Chun (Jack) Huang, for the gift of wild-type, W1118, *Drosophila* flies and for maintaining our mutant fly stocks. We also thank Prof. James Vigoreaux (University of Vermont) for the gift of the Dmlc2[Δ2-46; S66A,S67A] fly strain and Prof. Donald LD Caspar for his insightful comments throughout this process. This research was funded by National Institutes of Health grant R01 GM030598 to KA Taylor and R00 HL124041 to MJ Previs. The Titan-Krios was partially funded by National Institutes of Health grant S10 RR025080 to KA Taylor. The DE-64 was purchased from funds provided by National Institutes of Health grant U24 GM116788 to the Southeastern Consortium for Microscopy of MacroMolecular Machines.

### Author Contributions

N Daneshparvar: data curation, formal analysis, validation, investigation, visualization, methodology, writing—original draft, review, and editing—analyzed cryoEM data, computed the reconstruction, and wrote the paper.
DW Taylor: resources, formal analysis, methodology, writing—original draft—and prepared the specimen.
TS O'Leary: formal analysis, investigation, and collected and analyzed mass spectrometry data.

H Rahmani: formal analysis, validation, investigation, methodology, and contributed to the data processing.

F Abbasiyeganeh: formal analysis, investigation, methodology, and cryoEM data collection.

MJ Previs: data curation, formal analysis, supervision, funding acquisition, investigation, writing—original draft, review, and editing—and analyzed mass spectrometry data.

KA Taylor: conceptualization, supervision, funding acquisition, and writing—original draft, review, and editing.

## Conflict of Interest Statement

The authors declare that they have no conflict of interest.

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
