## [Reviewer comments · Life Science Alliance]

CryoEM Structure of *Drosophila* Flight Muscle Thick Filaments at 7Å Resolution

Nadia Daneshparvar, Dianne Taylor, Thomas O'Leary, Hamidreza Rahmani, Fatemeh Abbasiyeganeh, Michael Previs, and Kenneth Taylor
DOI: 10.26508/lsa/202000823

Corresponding author(s): Prof. Kenneth A. Taylor (Institute of Molecular Biophysics, Florida State University)

Review timeline:

Submission Date:	2020-06-22
Editorial Decision:	2020-06-24
Revision Received:	2020-06-26
Editorial Decision:	2020-06-29
Revision Received:	2020-06-30
Accepted:	2020-06-30

Transaction Report:

No Peer Review Process File is available with this article, as the authors have chosen not to make the review process public in this case.

RE: Life Science Alliance Manuscript #LSA-2020-00823-T

Prof Kenneth A. Taylor
Florida State University
Institute for Molecular Biophysics
Off Chieftain Way
Tallahassee, FL 32306-4380

Dear Dr. Taylor,

Thank you for submitting your revised manuscript entitled "CryoEM Structure of Drosophila Flight Muscle Thick Filaments at 7Å Resolution". We would be happy to publish your paper in Life Science Alliance pending final revisions necessary to meet our formatting guidelines.

- please have all corresponding authors add their ORCID ID - you should have received instructions on how to do so
- please add all required information to our system (summary blurb, category, author contributions, etc.)
- please add a conflict of interest statement to the main manuscript text
- please add all figures in separate files, including the main figures and the supplementary figures (for more information on figure formatting, please look at our author guidelines)
- please add your figure legends for main figures, supp. Figures, and movies to the main manuscript text
- please provide your manuscript and tables in editable docx or excel format versions
- please list 10 authors and et al. in the reference list
- please take a look at your figure callouts once more - on page 11, there is a callout for Figure 5, but the manuscript does not have a Figure 5
- Please add information on p-values and number of replicates in the figure legends where appropriate
- Please limit keywords to maximally five
- please consult our Manuscript Preparation Author Guidelines and follow the instructions of how the sections of your manuscript should be divided
- Remove separate 'Statement of Significance' and 'Conclusions' paragraphs.
- Rename 'Data depositions' section to 'Data availability' and update the accession codes.
- Introduce a separate 'Statistical Analysis' paragraph within the Material and Methods.

A. FINAL FILES:

B. MANUSCRIPT ORGANIZATION AND FORMATTING:

Sincerely,

Reilly Lorenz
Editorial Office Life Science Alliance
Meyerhofstr. 1
69117 Heidelberg, Germany
t +49 6221 8891 414
e contact@life-science-alliance.org
www.life-science-alliance.org

RE: Life Science Alliance Manuscript #LSA-2020-00823-TR

Prof. Kenneth A. Taylor
Florida State University
Institute of Molecular Biophysics
91 Chieftan Way
Tallahassee, FL 32306-4380
United States

Dear Dr. Taylor,

Thank you for submitting your revised manuscript entitled "CryoEM Structure of Drosophila Flight Muscle Thick Filaments at 7Å Resolution". We would be happy to publish your paper in Life Science Alliance pending final revisions necessary to meet our formatting guidelines.

-please add the figure legends for Supplementary Figures, table legends, and movies to the main manuscript text

A. FINAL FILES:

B. MANUSCRIPT ORGANIZATION AND FORMATTING:

Sincerely,

Reilly Lorenz
Editorial Office Life Science Alliance
Meyershofstr. 1
69117 Heidelberg, Germany
t +49 6221 8891 414
e contact@life-science-alliance.org
www.life-science-alliance.org

RE: Life Science Alliance Manuscript #LSA-2020-00823-TRRR

Prof Kenneth A. Taylor
Florida State University
Institute for Molecular Biophysics
Off Chieftain Way
Tallahassee, FL 32306-4380

Dear Dr. Taylor,

Thank you for submitting your Research Article entitled "CryoEM Structure of Drosophila Flight Muscle Thick Filaments at 7Å Resolution". It is a pleasure to let you know that your manuscript is now accepted for publication in Life Science Alliance. Congratulations on this interesting work.

DISTRIBUTION OF MATERIALS:

Again, congratulations on a very nice paper. I hope you found the review process to be constructive and are pleased with how the manuscript was handled editorially. We look forward to future exciting submissions from your lab.

Sincerely,

Reilly Lorenz
Editorial Office Life Science Alliance
Meyerhofstr. 1
69117 Heidelberg, Germany

t +49 6221 8891 414
e contact@life-science-alliance.org
www.life-science-alliance.org